# Spatial and Temporal Variations of Predicting Fuel Load in Temperate Forests of Northeastern Mexico

**Ma. del Rosario Aradillas-González** [1]**, Virginia Vargas-Tristán** [2]**, Ausencio Azuara-Domínguez** [1]**,**
**Jorge Víctor Horta-Vega** [1]**, Javier Manjarrez** [3]**, Jorge Homero Rodríguez-Castro** [1]
**and Crystian Sadiel Venegas-Barrera** [1,*]

1   División de Estudios de Posgrado e Investigación, Instituto Tecnológico de Ciudad Victoria,
    Tecnológico Nacional de México, Boulevard Emilio Portes Gil, 1301, Ciudad Victoria P.C. 87010, Mexico;
    rossyara235@gmail.com (M.d.R.A.-G.); ausencio.ad@cdvictoria.tecnm.mx (A.A.-D.);
    jorge.hv@cdvictoria.tecnm.mx (J.V.H.-V.); jorge.rc@cdvictoria.tecnm.mx (J.H.R.-C.)
2   Facultad de Ingeniería y Ciencias, Centro Universitario Ciudad Victoria, Universidad Autónoma de Tamaulipas,
    Ciudad Victoria P.C. 87149, Mexico; vvargas15@hotmail.com
3   Centro de Investigación en Recursos Bióticos, Universidad Autónoma del Estado de México,
    Instituto Literario, Toluca P.C. 50000, Mexico; jsilva@uaemex.mx
*   Correspondence: crystian.vb@cdvictoria.tecnm.mx

**Abstract:** The prediction of fuel load areas and species associated with these events reduces the response time to fight forest fires. The objective of this study was to estimate the annual fuel load from 2009–2013, predict the annual fuel load in the rest of the ecosystem, identify species that contribute most to this load and compare the percentage of area by risk category in the temperate forests of Tamaulipas. Fuel load was estimated with inventory data using three models. Fuel load was predicted with elevation, total annual precipitation, mean annual temperature, and enhanced vegetation index from satellite scenes using partial least squares regression. The highest concentration of fuel load was associated with the oak, oak-pine, pine forest and mountain mesophyll forest ecosystems. The contribution of genera to fuel load was different. *Quercus* contributed the most variation among clusters, and the contribution among *Quercus* species was similar. The results highlight the importance of focusing fuel management programs on this type of ecosystem, emphasizing actions in particular *Quercus*, and the results can also serve as a basis for future research, such as carbon sequestration and forest management programs.

**Keywords:** forest fire; clusters; vegetation index; partial least squares; PERMANOVA

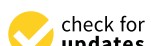



## 1. Introduction

Fuel load estimation allows the prediction and identification of forest fire risk areas [1,2]. Forest fires affect approximately 34 million ha/year worldwide [3]. Fuel load can be estimated from direct measurements [4], by allometric equations [5–7] and by expansion factors, which transform the volume of forest per tree or per unit area into fuel load density in m³/ha [8–10]. Point estimates are useful for predicting fuel load in areas where measurements are lacking [11]. Predictions are made by mathematical models using independent environmental variables, such as topography [12], vegetation, and climatic indices [13–17]. The models represent a means to reduce the economic and logistical costs associated with their estimation and in combination with field data help to reduce the bias in fuel load estimation [18,19]. However, factors that determine fuel load vary over time; therefore, it is necessary to include variables that reflect the variation, such as vegetation indices. Vegetation indices are derived from satellite images, which express the reflectance of the red and near-infrared electromagnetic regions [20].

Vegetation indices, derived from satellite imagery, reflect spatiotemporal variations in primary productivity and allow the characterization of different vegetation covers [21,22].

Among the most used indices are the normalized green–red difference index, ground-adjusted vegetation index and enhanced vegetation index (EVI). The normalized green–red difference index is widely used, but it is affected by the presence of aerosols in the atmosphere and canopy interference and is less sensitive to dense areas [23,24]. Therefore, it is necessary to explore the use of other indices that reduce these biases, such as the EVI, which is used to predict fuel load and fuel moisture and evaluate forest cover [25,26]. The advantage of these layers is that they are available from 2000 to the present, have a periodicity of 16 days and are freely available [20,27]. The EVI is sensitive to regions of high biomass, such as those generated by temperate forests [28].

Temperate forest ecosystems, which cover approximately 10 million square kilometers in the world [29], occupy third place in net primary productivity [30], and from them, it is possible to obtain parameters of vegetation cover, such as density, leaf area index and chlorophyll activity, to estimate fuel load and to generate fire severity models [31–35]. Mexico's temperate forests cover approximately 16.5% of its surface and are a priority for conservation. The temperate forests of Mexico represent one of the three global hotspots, regions characterized by their high endemism of plants that survive in 30% or less of their original distribution [36]. The Madrean pine-oak forests contain nearly 5,300 species of flowering plants and have a high diversity of pines, with 44 of the 110 recognized species of pines, more than 135 species of oaks and more than 30% of the species of this genus in the world, 85 of which are endemic to Mexico [37]. In addition, it is an important source of environmental services due to its high degree of endemism and species diversity [38], such as pines, shrubs, bromeliads, and orchids [39]. However, these ecosystems face constant forest fires that in the last 16 decades have affected 7 million ha of temperate and boreal forests [3]. In the state of Tamaulipas, 40% of fires occur in temperate forests [40]. Temperate forests represent 7.21% of the state's native surface, of which 28.3% are coniferous and broadleaf species [41]. The temperate forests of Tamaulipas occupy 524 thousand ha, of which approximately 27% were conserved in the biosphere reserve "El Cielo" [42]; due to the importance of the ecosystem, it is necessary to generate fuel load maps that identify temporal and spatial changes.

In this study we predict fuel load from three models to select the optimal method in the temperate forests of Tamaulipas, identify temporal variations among ecosystem types, and identify forest genera and species that contribute most to fuel load in the study area. We propose that the improved vegetation index increases the predictive capacity of fuel load than variables that do not change over long periods of time such as topography, altitude, and orientation. Determining the fuel load in these ecosystems is fundamental for continuing the conservation and use of temperate forests such as the reduction of the fuel load.

## 2. Materials and Methods

### 2.1. Study Area

The state of Tamaulipas is localized in the northeastern part of Mexico, bordered to the north by the United States of America, to the east by the Gulf of Mexico, to the south by the state of Veracruz, to the southwest by San Luis Potosí and to the west by Nuevo León (Figure 1). The temperate forests of Tamaulipas are found in Sierra Madre Oriental, southwest of the state, and in part of the Northern Gulf Coastal Plains physiographic province, which extends from sea level to a 4000 m elevation [43]. The temperate forests of Tamaulipas cover an area of 524,000 ha and comprise the largest area of oak forest, followed by oak-pine, cloud forest, pine-oak forest and tascate forest [44].

### 2.2. Description of the Study Area

The climate corresponds to the temperate (C) group, but the semi-warm subhumid (A)C(wo) subtypes dominate, characterized by an average annual temperature of 18 °C, a temperature of the coldest month lower than 18 °C, and a temperature of the hottest month higher than 22 °C. The precipitation of the driest month is less than 40 mm, summer

rainfall has a total precipitation index less than 43.2, and the percentage of winter rainfall ranges from 5% to 10.2% of the annual total. The dominant soil units are lithosols, vertisols and redzins.

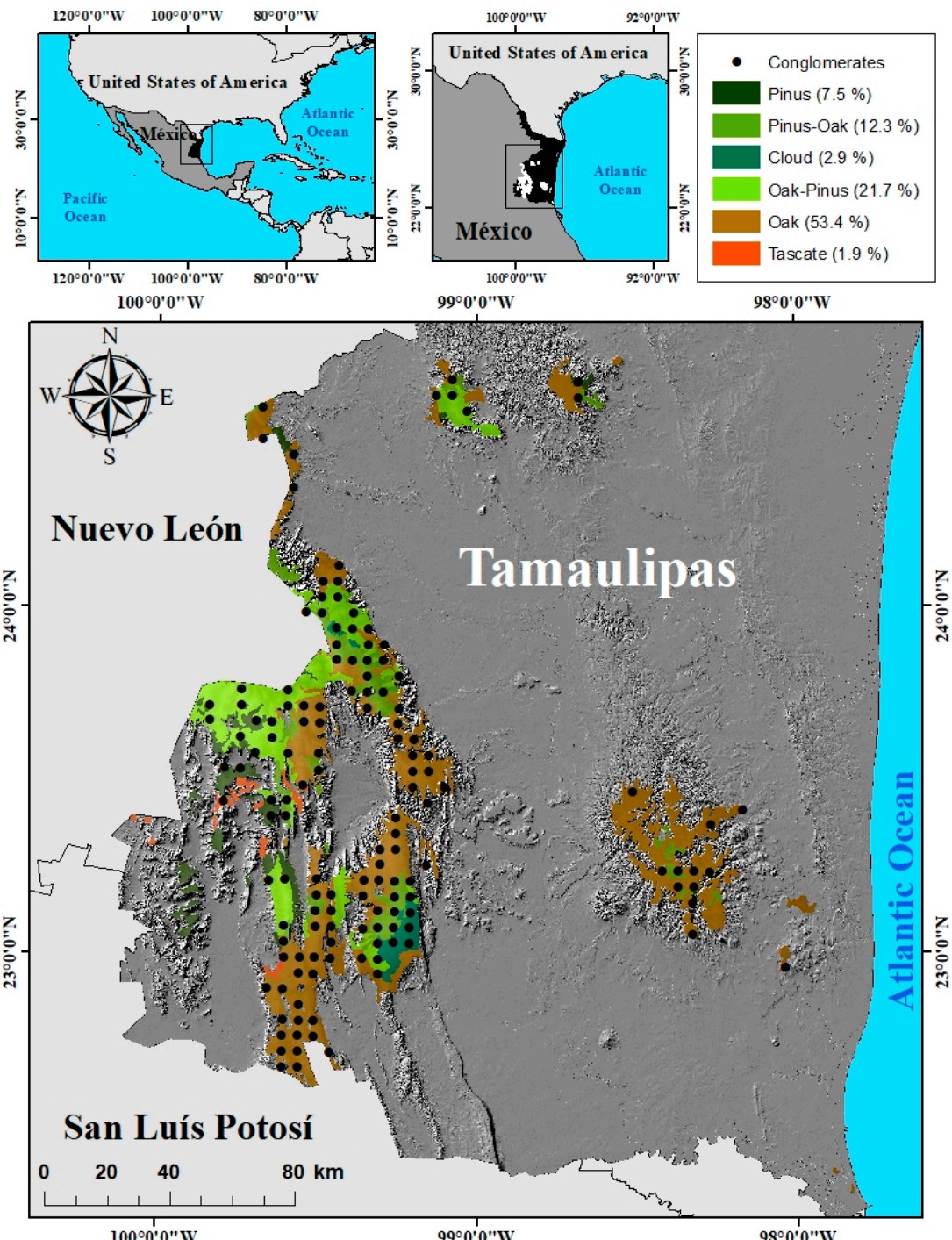

**Figure 1.** Location of the study area and conglomerates used to estimate fuel load in SE, central and coastal Gulf of Mexico, Tamaulipas, Mexico.

*2.3. Database*

The fuel load was estimated with data from the State Forest and Soil Inventory 2009–2014 (IEFyS, 2009–2014) [42]. We evaluated 141 conglomerates obtained from Tamaulipan forests from 2009 to 2013 (Figure 1). The conglomerates evaluated consist of four sampling subsites of 400 m$^2$ of woodland, giving an area of 1600 m$^2$. The IFEyS method fol-

lowed a systematic stratified sampling design, which was separated by a linear equidistance of 2.5 × 2.5 km. The conglomerate consisted of two sampling units, a primary sampling unit (PSU) and four secondary sampling units (SSUs). The conglomerate consisted of a one-hectare circular plot (56.42 m radius), and the four secondary sampling units (SSUs), or sites, were separated at 45.14 m, arranged in a north-facing inverted Y-shape (Figure 2). The tree canopy variables recorded were diameter, height, and canopy cover.

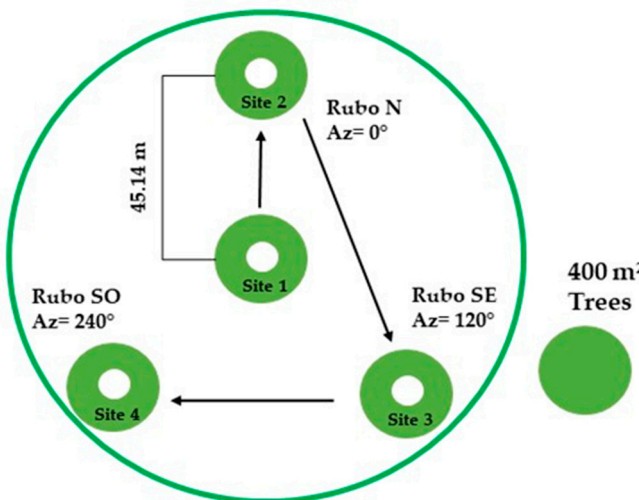

**Figure 2.** Conglomerates and sampling sites in the forest area. Data are from the State Forest and Soil Inventory of Tamaulipas 2009–2014.

*2.4. Fuel Load Estimation*

The fuel load per conglomerate was estimated with three models for temperate forests using the aboveground biomass of *Quercus*, *Pinus*, *Arbutus*, *Juniperus*, *Juniperus*, *Liquidambar*, *Cedrela*, *Carpinus*, *Cupressus*, *Conestegia*, *Cercocarpus* and *Sargentia* genus. Model 1 was the volume equation for temperate forest species, which considers the total real stock or volume per hectare [9]. Model 2 included equations to estimate the aboveground biomass of five species of *Quercus*, four species of *Pinus*, *Juniperus flaccida* and broadleaved species that included the normal diameter (DN), height in cm, a constant and a weighting exponent by species [17,45,46] as parameters. Model 3 used the equation designed for coniferous species involving normal diameter (DN), constant value and a weighting exponent [47] (Table 1). The models consider different parameters; therefore, it is necessary to identify the best model for the temperate forests of Tamaulipas.

*2.5. Fuel Load Comparison between Models*

The three models to estimate fuel load to temperate forest use different variables; therefore, we compared them to identify the differences in fuel load estimation. We compare the fuel load of each conglomerate obtained from the three models, and since each conglomerate has three estimates, we use a student's *t*-pair test with an *α* of 0.05. This method tests whether the mean of sample differences between pairs of models is significantly different from zero [48].

*2.6. Prediction Fuel Load*

The fuel load was predicted with topographic (slope and elevation), climatological (annual evapotranspiration, annual total precipitation, and annual average temperature), vegetation (canopy height) and the enhanced vegetation index from February to September (2008 to 2014) (Table 2). The EVI was obtained from MOD13Q1 imagery at a temporal resolution of 16 days and a spatial resolution of 250 m.

The EVI reduces both atmospheric interference and soil saturation, is more sensitive to canopy structural variations and is more reliable under high-biomass conditions [49].

The index incorporates an "L" value to adjust for canopy background and a "C" value to compensate for atmospheric drag and blue band B values [50]. The relationship of the EVI to fuel load derived from Terra satellite scenes dated February to September coincided with the period when field sampling was conducted. The layers were rescaled from a cell size of 250 m to an arcsecond. The images were projected to Universal Transverse Mercator Zone 14N.

**Table 1.** Equations of three models used to estimate fuel load.

| Model | Species | Equation | Source |
|---|---|---|---|
| Model real stock | Temperate forest | Vt = α × DB1 × HB2 | Silva-Arredondo y Návar-Cháidez, (2009) |
| Model aboveground biomass | *Quercus cambyi* | B = α − 2.3112 × D2.4497 | Rodríguez-Laguna et al. (2007) |
| | *Quercus laceyi* | B = α − 2.4344 × D2.5069 | Rodríguez-Laguna et al. (2007) |
| | *Quercus rysophylla* | B = α − 2.2089 × D2.3736 | Rodríguez-Laguna et al. (2007) |
| | *Quercus rugosa* | B = 0.089 × DN2.5226 | Rojas-García et al. (2015) |
| | *Quercus ssp* | B = 0.45534 × DN2 | Rojas-García et al. (2015) |
| | *Pinus greggi* | B = −0.177 + (0.015 × DN2) × h) | Rojas-García et al. (2015) |
| | *Pinus moctezumae* | B= 1.30454 × DN2.3644444 | Rojas-García et al. (2015) |
| | *Pinus nelsonni* | B = 0.1229 × DN2.3964 | Rojas-García et al. (2015) |
| | *Pinus teocote* | B = 0.032495 × DN2.766578 | Rojas-García et al. (2015) |
| | *Juniperus flaccida* | B = α − 1.6469 × DN2.1255 | Rodríguez-Luna et al. (2007) |
| | Broadleaf forest | B = EXP(−B0) × (DN2 ∗ h) B1 | Soriano-Luna et al. (2015) |
| Model coniferous species | Coniferous forest | B = 5.0 + 150000 × DN − 2.7/DN − 2.7 + 364946 | Brown et al. (1997) |

**Table 2.** Variables used to predict fuel load in the Tamaulipan temperate forest.

| Variable | Source | Accessed Date |
|---|---|---|
| Vegetation height (m) | http://www.earthenv.org | |
| Evapotranspiration | https://worldclim.org | |
| EVI | https://earthexplorer.usgs.gov | |
| Elevation (msnm) | https://www.inegi.org.mx | 5 April 2020 |
| Slope % | https://www.inegi.org.mx | |
| Annual precipitation (mm) | https://worldclim.org | |
| Annual average temperature (°C) | https://worldclim.org | |

The variables considered to predict the fuel load were 118 variables, which included vegetation height, evapotranspiration, the EVI of 112 satellite images, elevation, slope, total precipitation, and average temperature. We used the variance inflator factor (VIF = $1/(1 − r^2)$) as a criterion to exclude redundant variation among variables. We applied a multiple regression, where variables with higher correlation coefficients were used as predictor variables and the rest of the variables were used as independent variables. We excluded dependent variables if the VIF was greater than 5.0 because variation in this variable was contained in the rest of the independent variables. The procedure was repeated until no variable had a VIF higher than 5. The variables retained to predict fuel load were elevation, total annual precipitation, average annual temperature, and average and standard deviation of the EVI of the rainiest months from 2009–2014.

### 2.6.1. Prediction Models

The prediction of fuel load in the temperate forests of Tamaulipas was performed for each year. In 2009, we used 29 conglomerates, 22 conglomerates for 2010, 30 for 2011, 31 for 2012 and 29 for 2013. Seventy-five percent of the conglomerates were used to generate the model, and the remaining 25% were used to validate the model [51]. The choice of conglomerates to generate the model and validate it was random, using a random function in the Excel program. Since the choice of conglomerates was random, 10 models per year were generated. The environmental variables were correlated with the fuel load of the

partial least square regression model with factor analysis in the STATISTICA Version 13 program [52]. The CMP model is a prediction technique that combines the features of principal component analysis and multiple regression; the model has fewer restrictions than other multivariate multiple linear regression multivariate models [53]. The CMP algorithm extracted a set of latent factors that efficiently explained the covariance between the dependent and independent variables [54]

The generic equation included the effect of each variable, where the number of interactions was equal to the number of independent variables used, in this case five. Thus, the interactions used were single, double, triple, quadruple, and quintuple.

$$Y = b0 + b \times S + b \times D + b \times T + b \times C + b \times Q \qquad (1)$$

wherein:

Y = dependent variable (fuel load obtained from models)
b = coefficient
S = single variable
D = double interaction of variables
T = triple interaction of variables
C = quadruple interaction of variables
Q = quintuple to interaction of variables

The models retained to predict fuel load were those that had a correlation equal to or greater than 0.5 between those obtained from models and those expected to be obtained from CMP. The coefficients of the retained models were averaged to generate the final prediction model. The average model was used to predict the fuel load.

### 2.6.2. Model Validation

Model validation was performed using two methods: regression and comparison of means between observed and obtained fuel loads on conglomerates. The comparison of the fuel load estimation models was performed through Student's *t*-test for paired samples with $\alpha = 0.05$. Student's *t*-test evaluates the significant difference between the means of two groups, where the hypothesis to test is the absence of differences between the mean values of the three models used for fuel load prediction [55]. Twenty-five percent of the clusters were used to validate the models. The validation consisted of comparing the estimated values with respect to the values obtained with the prediction. The comparison was performed with Student's *t*-test for independent samples, which compares the mean of the two groups; the data used to validate the model were randomly assigned. The analysis was performed with the STATISTICA program [52].

### 2.7. Genus and Species Contribution

This study compared the number of individuals by genus and species among three fuel load categories through multivariate PERMANOVA using Past [56]. Fuel load was classified into three risk categories, taking as a reference the fuel behavior model, which separates the load into low (0 to 13 Mg ha$^{-1}$), medium (13.1 to 23 Mg ha$^{-1}$), and high ($23.1 \geq$ Mg ha$^{-1}$) [57]. PERMANOVA is a variant of the ANOVA test, which compares multidimensional distance within and between groups [58]. The Bray–Curtis analysis compared the specific composition within and between 12 genera, as well as between 31 tree species, under the null hypothesis that there is no difference between the structure (number of individuals per genus) of the three risk categories [59]. The percentage contribution of species was determined with the SIMPER test to determine the difference or similarity between groups [60].

### 2.8. Temporal Variations in Fuel Load Categories by Type of Forest

The temporal variations (years) by fuel load categories were determined with a simple correspondence test. The year variable was composed of five categories and fuel risk with

three categories (low, medium, and high) and areas with clouds. The simple correspondence test is a modification of the $X^2$ test that measures the association of the categories of two categorical variables and plots them on a two-dimensional graph based on the association between the categories [61]. The position of the categories on the graph represents the degree of association between categories of different variables. The interpretation of the graph is based on the canonical position of categories on the biplot graph. When two categories of different variables were near, it was associated with a high percentage of area in a fuel load category and a specific year; otherwise, if the positions of two categories were separated, then there was a low proportion of area per fuel load category in the year evaluated. The fuel load categories and years were ranked according to the percentage in the area in which each fuel load category occurred in each year. The types of vegetation present were pine forest, tascate forest, pine-oak forest, cloud forest, oak-pine forest, and oak-oak forest [42].

## 3. Results

### 3.1. Fuel Load

The estimated fuel load was similar among the three models in the years evaluated. Real stock presented the highest estimated fuel load values, where the estimated fuel load was statistically similar for 2009 and 2013 (Table 3). The fuel load estimation among the three models was statistically similar. The estimation with aboveground biomass and coniferous species models was statistically similar for 2009, 2010, 2011, but different in 2012. Real stock and aboveground biomass estimates were similar. The real stock and coniferous species models were different in 2010 and 2011 (Table 4).

**Table 3.** Comparison of fuel load prediction models by year.

| Year | Model | $\bar{x}$ | s | Dif $\bar{x}$ | *t* Value | d.f | *P* |
|---|---|---|---|---|---|---|---|
| 2009 | Real stock<br>Aboveground biomass | 17.1<br>13.4 | 21.3<br>16.7 | 3.7 | 0.7 | 28.0 | 0.483 |
| | Real stock<br>Coniferous species | 17.1<br>13.4 | 21.3<br>16.4 | 3.6 | 0.7 | 28.0 | 0.458 |
| | Aboveground biomass<br>Coniferous species | 13.4<br>13.4 | 16.7<br>16.4 | 0.0 | 0.0 | 28.0 | 0.981 |
| 2010 | Real stock<br>Aboveground biomass | 17.1<br>10.9 | 9.9<br>11.9 | 6.1 | 1.7 | 21.0 | 0.097 |
| | Real stock<br>Coniferous species | 17.1<br>10.6 | 9.9<br>10.8 | 6.5 | 2.4 | 21.0 | 0.024 |
| | Aboveground biomass<br>Coniferous species | 10.9<br>10.6 | 11.9<br>10.8 | 0.4 | 0.1 | 21.0 | 0.915 |
| 2011 | Real stock<br>Aboveground biomass | 18.4<br>15.7 | 14.1<br>14.1 | 2.7 | 0.977 | 29.0 | 0.336 |
| | Real stock<br>Coniferous species | 18.4<br>13.65 | 14.1<br>11.8 | 4.7 | 6.9 | 29.0 | 0.000 |
| | Aboveground biomass<br>Coniferous species | 15.7<br>13.7 | 14.1<br>11.9 | 2.0 | 0.7 | 29.0 | 0.432 |
| 2012 | Real stock<br>Aboveground biomass | 19.3<br>16.7 | 15.9<br>12.9 | 2.5 | 1.5 | 30.0 | 0.129 |
| | Real stock<br>Coniferous species | 19.3<br>13.8 | 15.9<br>11.1 | 5.5 | 3.2 | 30.0 | 0.029 |
| | Aboveground biomass<br>Coniferous species | 16.7<br>13.8 | 12.9<br>11.1 | 3.0 | 5.9 | 30.0 | 0.000 |

**Table 3.** *Cont.*

| Year | Model | x̄ | s | Dif x̄ | *t* Value | d.f | *P* |
|------|-------|------|------|------|------|------|------|
| 2013 | Real stock<br>Aboveground biomass | 16.7<br>11.3 | 26.2<br>26.4 | 5.4 | 0.8 | 28.0 | 0.427 |
| | Real stock<br>Coniferous species | 16.7<br>9.7 | 26.2<br>27.3 | 7.0 | 1.1 | 28.0 | 0.278 |
| | Aboveground biomass<br>Coniferous species | 11.3<br>9.7 | 26.4<br>27.3 | 1.6 | 0.2 | 28.0 | 0.833 |

x̄ = average, s = standard deviation, Dif. x̄ = average differences, *t* value = value of paired *t* test, d.f. = degree of freedom, and *P* = probability.

**Table 4.** Analyses of validation results for 25% of the conglomerates in the three prediction models by year.

| Year | Model | Data | x̄ | s | Dif. x̄ | *t* Value | d.f | *P* |
|------|-------|------|------|------|------|------|------|------|
| 2009 | Real stock | Observed<br>Predicted | 20.27<br>17.13 | 17.00<br>30.27 | 3.14 | 0.45 | 7 | 0.67 |
| | Aboveground biomass | Observed<br>Predicted | 20.27<br>9.12 | 17.00<br>23.99 | 11.15 | 1.63 | 7 | 0.15 |
| | Coniferous species | Observed<br>Predicted | 20.27<br>9.97 | 17.00<br>19.27 | 10.30 | 1.56 | 7 | 0.16 |
| 2010 | Real stock | Observed<br>Predicted | 20.21<br>17.83 | 7.90<br>8.69 | 2.37 | 1.55 | 8 | 0.16 |
| | Aboveground biomass | Observed<br>Predicted | 17.26<br>14.82 | 7.66<br>7.71 | 2.44 | 1.83 | 8 | 0.11 |
| | Coniferous species | Observed<br>Predicted | 15.18<br>14.06 | 5.56<br>4.50 | 1.13 | 1.08 | 8 | 0.31 |
| 2011 | Real stock | Observed<br>Predicted | 17.70<br>29.01 | 13.63<br>13.80 | −11.31 | −5.90 | 6 | 0.001 |
| | Aboveground biomass | Observed<br>Predicted | 18.31<br>23.97 | 13.70<br>12.45 | −5.66 | −1.48 | 6 | 0.19 |
| | Coniferous species | Observed<br>Predicted | 18.31<br>20.80 | 13.70<br>11.02 | −2.49 | −1.20 | 6 | 0.27 |
| 2012 | Real stock | Observed<br>Predicted | 17.89<br>26.82 | 11.01<br>19.68 | −8.93 | −1.15 | 7 | 0.29 |
| | Aboveground biomass | Observed<br>Predicted | 17.89<br>22.32 | 11.01<br>8.25 | −4.43 | −0.92 | 7 | 0.39 |
| | Coniferous species | Observed<br>Predicted | 13.03<br>18.91 | 8.35<br>6.12 | −5.88 | −1.62 | 7 | 0.15 |
| 2013 | Real stock | Observed<br>Predicted | 13.07<br>14.20 | 17.01<br>26.45 | −1.13 | −0.26 | 6 | 0.81 |
| | Aboveground biomass | Observed<br>Predicted | 11.60<br>12.24 | 14.81<br>20.31 | −0.65 | −0.24 | 6 | 0.82 |
| | Coniferous species | Observed<br>Predicted | 9.57<br>10.43 | 12.98<br>20.42 | −0.86 | −0.24 | 6 | 0.82 |

x̄ = average, s = standard deviation, Dif. x̄ = average differences, *t* value = value of paired *t* test, d.f. = degree of freedom, and *P* = probability.

### 3.2. Prediction

The correlation coefficient ($r^2$) between the conglomerate estimated fuel load and that obtained by prediction with the partial least square regression model varied from 0.54 (coniferous species model of 2009), while the highest values were presented in the real

stock and coniferous species models of 2011 and the real stock model of 2013 (Figure 3). Therefore, predictions had a medium- to high-predictive capacity to predict fuel load on temperate forests. The highest fuel load was in areas of higher elevation in the eastern part of the state, as well as in the Sierra Madre Oriental, Sierra de San Carlos, and Sierra de Tamaulipas (Figure 4).

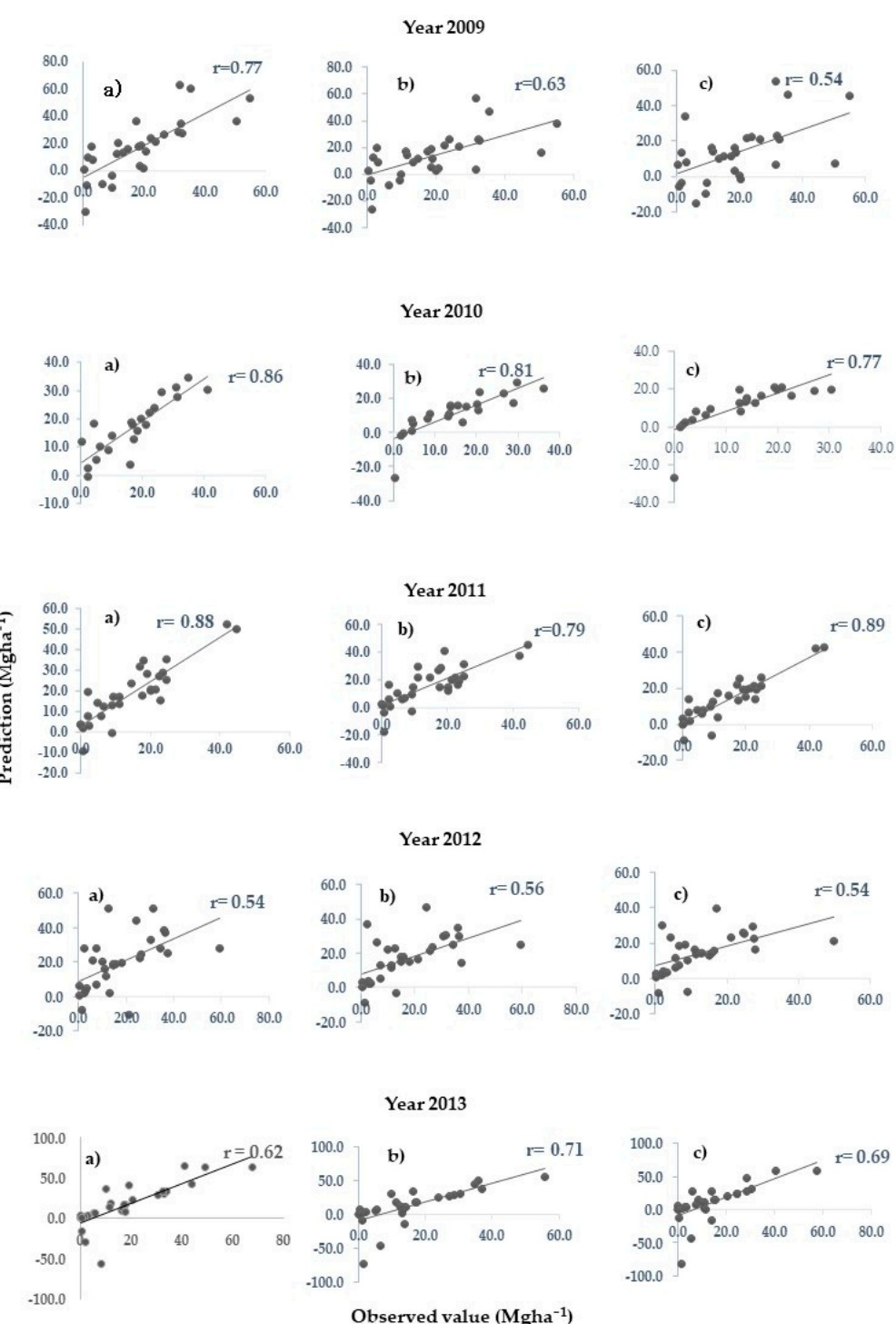

**Figure 3.** Relationship between estimated fuel load in conglomerates and that obtained by prediction with the partial least squares regression model with CMP factor analysis for the models. Subfigures (**a**) real stock, (**b**) aboveground biomass; and (**c**) coniferous species.

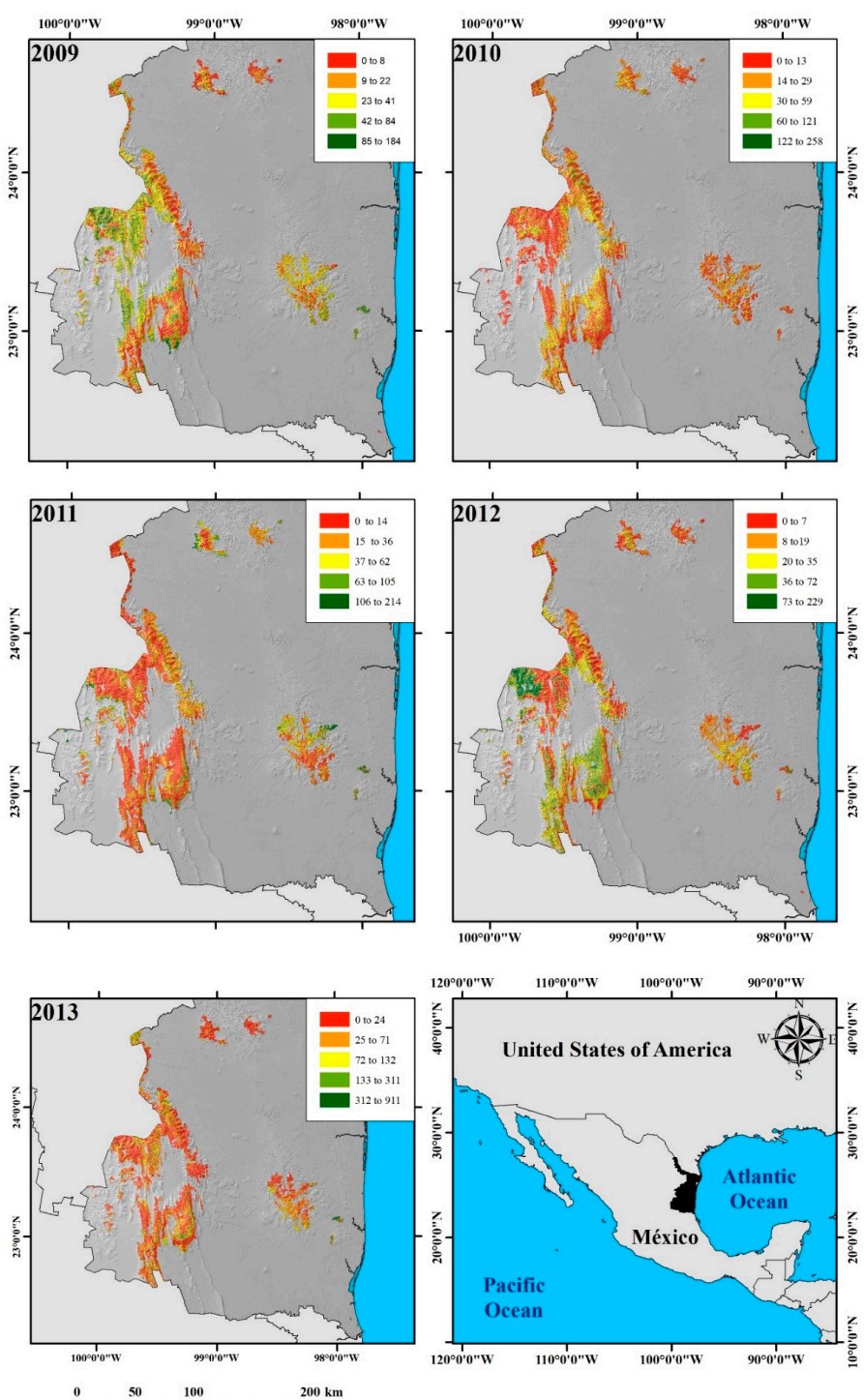

**Figure 4.** Maps of predicted average fuel load in the temperate forest ecosystem for 2009, 2010, 2011, 2012, and 2013 obtained in Tamaulipas, Mexico.

The estimated fuel load values were statistically like those predicted with the CMPs, except for real stock from 2011, which showed differences between observed and expected values (Table 4).

### 3.3. Contribution of Genera and Species to Fuel Load

The fuel load categories presented significant differences in plant structure (PERMANOVA, F = 6.48, *p* > 0.0001). Differences occurred between three fuel load categories,

high and medium (F = 2.9, *p* > 0.0001), high and low (F = 8.5, *p* > 0.02), and medium and high (F = 2.9, *p* > 0.02), as well as between low and medium (F = 5.6, *p* > 0.007). *Quercus* contributed the greatest differences among the three categories; the frequency of individuals of the genus was the highest in the risk category, followed by the medium-risk category, and the lowest frequency of individuals was recorded in the low-risk category (Table 5). Species of *Pinus* contributed the second highest variation among the three fuel load categories. The contribution analysis of the 31 species of *Quercus* was similar among the risk intervals (F = 0.88 and *p* = 0.70).

**Table 5.** Genus's contribution.

| Genus | Contribution (%) | Acumulative Contribution (%) | Risk Forest Fire | | |
| --- | --- | --- | --- | --- | --- |
| | | | High | Medium | Low |
| *Quercus* | 68.7 | 68.7 | 66.7 | 57.2 | 31.1 |
| *Pinus* | 17 | 85.7 | 11.9 | 2.8 | 3.1 |
| *Arbutus* | 6.2 | 91.9 | 3.6 | 3.1 | 1.8 |
| *Juniperus* | 3.5 | 95.4 | 3.8 | 2 | 0.6 |
| *Liquidambar* | 2.6 | 98 | 1.7 | 0.9 | 0.1 |
| *Cedrela* | 0.5 | 98.5 | 0.2 | 0.1 | 0.1 |
| *Carpinus* | 0.5 | 99.1 | 0.3 | 0 | 0.1 |
| *Cupressus* | 0.4 | 99.4 | 0.1 | 0 | 0.1 |
| *Conestegia* | 0.2 | 99.7 | 0.1 | 0 | 0 |
| *Cercocarpus* | 0.2 | 99.9 | 0 | 0 | 0.2 |
| *Randia* | 0.1 | 100 | 0 | 0 | 0.1 |
| *Sargentia* | 0 | 100 | 0 | 0 | 0 |

### 3.4. Variation in Fuel Load by Risk Forest Fires

The percentage of area by risk category varied between years, and medium- and low-risk fire loads were more frequent in oak and pine-oak forest in 2009. Clouds were more frequent in pine and cloud forest in 2011 and 2013. A high category of fuel load, oak-pine and tascate, as well as 2012 and 2010, were not associated with a particular year, type of forest or category of fuel load (Figure 5).

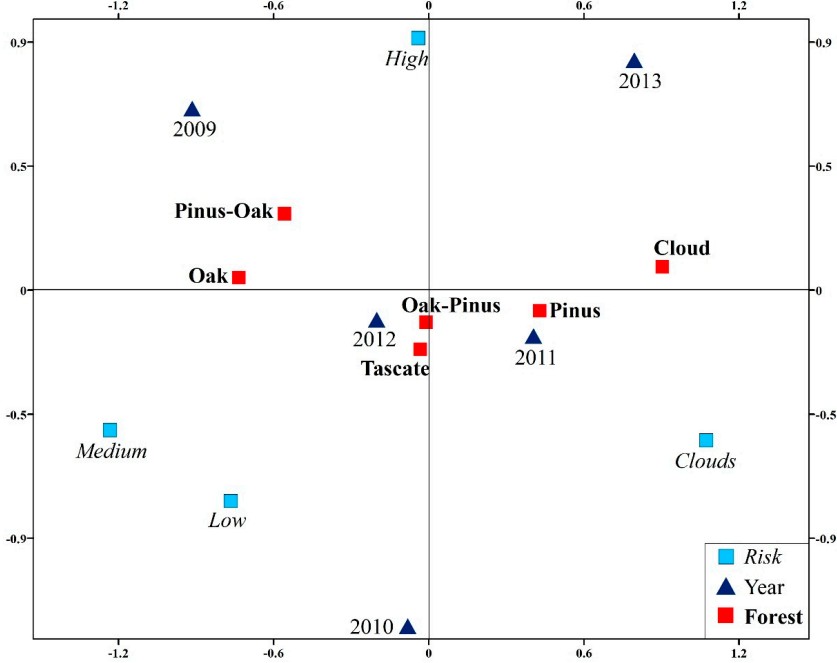

**Figure 5.** Association of three risk categories of fuel load and clouds (blue squares), type of temperate forest (red diamonds) and year (blue navy triangles).

## 4. Discussion

This study estimated fuel load using inventory data through allometric, volume and temperate forest equations reported in the literature, grouped into three different models (Table 1). The real stock model could not predict fuel load in 2011 and tended to generate higher values, while the coniferous species models could predict fuel load in all years but tended to estimate lower values. Aboveground biomass was a model that predicted all years and estimated intermediate values. The real stock model underestimated the fuel load, while the coniferous species model overestimated the fuel load. Model 2 avoided overestimating or underestimating the fuel load in temperate forests. The aboveground biomass model included species equations for temperate forests, making it easier to estimate fuel load [17,45,46]. The use of species equations combined with field data helps to reduce prediction bias [62]. Although the models allowed the prediction of the fuel load within the temperate forest ecosystem in Tamaulipas, it is advisable to carry out measurements to contrast the results obtained in this research. The values reported in this study were like those reported by other authors for temperate forest ecosystems [9,17,33,44], which ranged from 0.54 Mg ha$^{-1}$ to 157.01 Mg ha$^{-1}$, by clusters and by species in pine forest, *Quercus* Forest, and mountain mesophyll forest. The values are similar to those reported by other authors in other parts of the world; for example, in Indonesian forests an average of 70 to/ha was estimated [63], in forests of southeastern Utah in the United States estimates ranged from 10 Mg ha$^{-1}$ to 100 Mg ha$^{-1}$ [64], in Chinese forests an average of 69 Mg ha$^{-1}$ was reported [65] and in eastern North Carolina an average of 20 Mg ha$^{-1}$ to 120 Mg ha$^{-1}$ was estimated [66]. The variables used in this study that contributed to predicting fuel load are related to primary productivity, such as the EVI. Vegetation indices reflect the spatiotemporal variation in primary productivity [21,49,67] and are better predictors than variables that remain constant over long periods of time, such as topography [12]. Vegetation indices, which can be freely obtained, are variables that can reduce economic and logistical costs in the evaluation of forestry resources. The uses of these variables include characterization of grasslands [33,68], phenological development of crops and forests [69,70], biomass estimation, carbon sequestration [71–73] and fire monitoring and recovery of burned areas [74–77], to generate fire severity models [34]. In the temperate forests of Tamaulipas, they contributed to predicting the fuel load, identifying the sites with the highest fuel load. The highest fuel load is in the highest elevation areas of the Sierra Madre Oriental and the eastern part of the study area, namely, areas that present the highest humidity coming from the Gulf of Mexico. The months of higher humidity are related to the greater photosynthetic period of the vegetation and greater accumulation of fuel material; characteristics that coincide with the forest fire season in Mexico [78]. However, EVI values may vary according to the season and vegetation type [79], and EVI values in different seasons should be considered to generate future fuel models. Additionally, clouds can reduce our capacities to predict fuel load due to their high reflectance. We found temporal variations in surface cover by area covered by clouds, where 2011 and 2013 were more frequent in clouds and *Pinus* forests, which can reduce our capacity to estimate the fuel load. However, the frequency of months of clouds in an area can be used as a variable to predict load fuel because it is an indicator of humidity and primary productivity.

The spatial and temporal variations in forest fuel load are necessary for planning forest management strategies as well as for modeling and predicting fire behavior [2,80]. In the present study, fuel load was classified into three risk categories for the occurrence of forest fires and could also be used for carbon sequestration programs among other uses. Fuel load and its characteristics are used as variables in forest fire prediction models [81–83] because they present information on vegetation change, structure, and characteristics in very short periods of time. Fuel load, as reported in the present study, can be incorporated into the national forest fire early warning system because it provides better information than the hot spots involved in the current system since the hot spot is detected after the occurrence of a forest fire [84]. The fuel load and its characteristics can also be incorporated

into forest fire prevention systems, such as the systems used in Canada [85] and the United States [86]. The fuel load could be considered for the current Forest Fire Danger Prediction System for Mexico, since this system involves the dry fuel variable [83], and the fuel load would provide information about the intensity of the forest fire.

Temporal variation in fuel load from 2009 to 2013 could be related to droughts, hurricanes, rainfall, and frost that occurred in the years evaluated, which are variables that can be included in the prediction models. The results obtained from the present study can be used to predict the areas with the highest fire risk, since areas with a high-fuel load tend to present fires of greater intensity [87]. Fuel load can also be used to estimate structural changes after a wildfire [88]. The temperate forests of Tamaulipas can present fires of great magnitude, since they exceed 23 Mg ha$^{-1}$ [57], which is the amount necessary to generate surface-to-crown fires [57]. The fuel load of higher plants influences the generation of litter and bark fuels. However, we recommend evaluating fuel load by type and stratum in temperate forests due to fuel load variations between types of forests. These estimates will allow for the evaluation of the evolution, direction and intensity of fires generated in forest ecosystems.

The present study highlights that fuel load was related to vegetation structure. Species of *Quercus* (oaks) contributed the greatest variation to fuel load, followed by oak-pine and oak-pine-oak forests. *Quercus* species are characterized by deciduous leaves, grouped on twigs. The trees can reach up to 20 m in height [89] and tend to be associated with different species of shrubs and herbaceous plants in the understory, in addition to numerous groups of epiphytes and mosses [39]. Species of *Quercus* present a wide plasticity in terms of resource arrangement, allowing them to adapt to a wide range of environments, which contributes to their great abundance and diversity [90,91]. Tamaulipas is one of the states with the highest species richness of *Quercus*, with 30% of the species reported nationally. However, it is necessary to use other models that include scrubs and herbaceous strata. The 0.5- to 2-m-high thickets and undergrowth in coniferous forests produce a fuel quantity of 10–15 t/ha (Rothermel Model 7) [92]. The contribution of *Quercus* to the fuel load in this research could be related to the high degree of association with other species, to its ecological functions such as the capacity to retain water and to being part of the habitat of other plant species [93].

*Quercus* presented the highest contribution to fuel load; therefore, in the temperate forests of Tamaulipas, fuel management operations oriented to this genus should be planned. In these ecosystems, *Quercus* species coexist with endemic species that are under special protection (*Brahea berlandieri*, *Brehea moori* and *Tilia mexicana*) and threatened species (*Beucarnea recurvata*, *Abies vejori* and *Carpinus caroliniana*) that are listed in NOM-059-SEMARNAT-2010 and that can be affected in the case of forest fires [44]. The prediction of fuel load and the species that contribute most to this parameter can be the subject of other studies, such as evaluating the health of the ecosystem, monitoring fires, creating a carbon inventory, prioritizing areas for fuel reduction and implementing forest management programs, and monitoring the structural changes of ecosystems before and after the occurrence of wildfires.

## 5. Conclusions

This study demonstrated the usefulness of the improved vegetation index (EVI) to predict the variation in fuel load in temperate forests of Tamaulipas. The aboveground biomass model allowed a model that can be used to predict fuel load. The results coincide with those reported by other authors in other forests around the world. The temperate forests of Tamaulipas present fuel loads $\leq$ 23 Mg ha$^{-1}$ capable of causing the different types of fires up to those of high intensity or crown fires. The highest concentration of fuel load was distributed in areas of higher productivity and elevation, located to the east of the temperate forest ecosystem, an area that receives moisture from the Gulf of Mexico and presents the highest occurrence of forest fires in the state. Pine, pine-oak and oak forests contain the highest fuel load. *Quercus* contributed significantly to the fuel load. The

results of this study can be considered a tool for the management of forest ecosystems and for the prevention of forest fires, forest stand monitoring, and continued evaluation of field data using other satellite data and models. They can also be considered a basis for future research.

**Author Contributions:** Conceptualization, M.d.R.A.-G., V.V.-T., A.A.-D., J.V.H.-V. and C.S.V.-B.; Data curation, M.d.R.A.-G., J.M. and J.H.R.-C.; Formal analysis, M.d.R.A.-G. and C.S.V.-B.; Investigation, M.d.R.A.-G., A.A.-D., J.M. and C.S.V.-B.; Methodology, M.d.R.A.-G., J.M., J.H.R.-C. and C.S.V.-B.; Project administration, C.S.V.-B.; Software, C.S.V.-B.; Validation, M.d.R.A.-G.; Visualization, M.d.R.A.-G., V.V.-T., A.A.-D., J.V.H.-V. and J.M.; Writing—original draft, M.d.R.A.-G., A.A.-D., J.V.H.-V., J.M., J.H.R.-C. and C.S.V.-B.; Writing—review and editing, M.d.R.A.-G., V.V.-T., A.A.-D., J.V.H.-V., J.M., J.H.R.-C. and C.S.V.-B. All authors have read and agreed to the published version of the manuscript.

**Funding:** M.d.R.A.-G. was supported by a scholarship of CONACyT (number 370000) to attend doctoral program in Sciences in Biology of the Technological Institute of Ciudad Victoria, Tamaulipas.

**Institutional Review Board Statement:** Not applicable.

**Informed Consent Statement:** Not applicable.

**Data Availability Statement:** Data are available from the National Forestry Commission (CONAFOR) and the authors' group and can be requested.

**Acknowledgments:** We thank to three reviewers for their comments that improved our manuscript.

**Conflicts of Interest:** The authors declare no conflict of interest.

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
