# Peer review of "Spatial and Temporal Variations of Predicting Fuel Load in Temperate Forests of Northeastern Mexico"

_forests, doi:10.3390/f13070988_

Round 1
Reviewer 1 Report
The article is well organized. It has all the necessary chapters. There is no need to supplement or shorten it.
Latin names of genera should always be written in italic.
The references in the list of references should be according to the author's guide.
The results of the research are very interesting and can be applied in practice.
Author Response
1.- Suggestion: “Latin names of genera should always be written in italic”
Authors: Pag. 14. L 361. “Quercus” = Quercus
Authors: The references list and citations style guide for MDPI Chicago journals v5 was used for the references and was complemented with v6. Pdf and the EndNote template for MDPI style was downloaded.
2.- Suggestion: “The references in the list of references should be according to the author's guide”.
Authors: changes were made to the following references: Pag. 15, Reference 3, L-427; Reference 6, L-434; Reference 9, L-440; Reference 15, L-445.
Pag. 16, Reference 20, L-467; Reference 29, L-487; Reference 33, L-495; Reference 34, L-498.
Pag. 17, Reference 60, L-551
Pag. 18, Reference 75, L-584.

Reviewer 2 Report
All comments and suggestions are in attached file.

Author Response
- Suggestion. It is recommended to the authors to conclude more clearly the role and importance of such research, and the possibility of applying such scientific research in the practical and actual situation in the field.
Authors: the research was concluded with the inclusion of recent studies related to the topic. Pag. 12, L-301-302.
Pag. 13, L-307-320; L-352-353.
Pag. 14, L-384-385; L-389-390; L-397-399.

Reviewer 3 Report
* I suggest developing the Introduction chap. also based on some projects results/official reports.
* The research questions should be more developed.
* Make sure to discuss your Conclusions in relation to other international studies hypotheses.
*The connection between results and some research projects could also be approached this could lead to an interesting and policy-relevant discussion
Author Response
Review 3
1.- Suggestion. I suggest developing the Introduction chap. also based on some projects results/official reports.
Authors: Pag. 1. L-39 to 41: Contribution of field data and prediction models in writing bias in fuel load estimation.
Pag 2. L-59: and to generate fire severity models
2.- Suggestion. The research questions should be more developed.
Authors: Research question was extended on Pag. 2. L-80 to 83: In the study it was proposed that the improved vegetation index increases the predictive capacity of fuel load than variables that do not change over long periods of time such as topography, altitude, and orientation. Determining the fuel load in these ecosystems is fundamental to continue the conservation, harvesting and management of temperate forests and to reduce the fuel load.
3.- Suggestion. Make sure to discuss your Conclusions in relation to other international studies hypotheses.
Authors: We include current data from other authors. Pag. 13. L-306-311. “The values are similar to those reported by other authors in other parts of the world, for example in Indonesian forests an average of 70 to/ha was estimated, in forests of southeastern Utah in the United States they estimated from 10 Mgha-1 to 100 Mgha-1, in Chinese forests an average of 69 Mgha-1 and in eastern North Carolina an average of 20 Mgha-1 to 120 Mgha-1 was estimated. L-320, to generate fire severity models. L-352-353. Fuel load can also be used to estimate structural changes after a wildfire.
Pag. 14. L-384-385. monitor the structural changes of ecosystems before and after the occurrence of wildfires
4.- Suggestion. The connection between results and some research projects could also be approached this could lead to an interesting and policy-relevant discussion
Authors: The results obtained in other studies in different parts of the world were discussed on Pag. 13 and 14 on L-306-311, L-320, L-352-353, L-384-385, L-357-359

Round 2
Reviewer 3 Report
All corrections have been made by the authors.